# Optimization of Process Parameters for ESR Waspaloy Superalloy by Numerical Simulation

**DOI:** 10.3390/ma15217483

**Published:** 2022-10-25

**Authors:** Jinguo Gao, Shulei Yang, Peng Zhao, Shufeng Yang, Jingshe Li, Wei Liu, Changle Zhang

**Affiliations:** 1School of Metallurgical and Ecological Engineering, University of Science and Technology Beijing, Beijing 100083, China; 2State Key Laboratory of Advanced Metallurgy, University of Science and Technology Beijing, Beijing 100083, China

**Keywords:** numerical simulation, electroslag remelting, waspaloy superalloy, molten pool, dendrite arm spacing

## Abstract

A transient numerical simulation method is used to investigate the temperature field, velocity field, and solidified field of large-size Waspaloy superalloy during the electroslag remelting (ESR) process. The effects of melting rate, filling rate, and thickness of the slag layer on the molten pool shape and dendrite arm spacing evolution have been discussed. The temperature in the slag pool is high and relatively uniformly distributed, the temperature range is 1690–1830 K. The highest temperature of the melt pool appears in the center of the slag–metal interface, 1686 K. There are two pairs of circulating vortices in the slag pool, the side vortices are caused by the density difference caused by the buoyancy of the slag, the center vortices are the result of the combined action of electromagnetic force and the momentum of the falling metal droplets. The molten pool depth and dendrite arm spacing increase with the increase of melting rate, but the slag layer thickness and electrode filling rate have little effect on the molten pool morphology and dendrite arm spacing if the droplet effect is not taken into account. Considering the morphology and depth of the molten pool as well as the size and distribution uniformity of the dendrite arm spacing, it is appropriate to maintain the melting rate at 5.8 kg/min for the industrial scale ESR process with the ingot diameter of 580 mm.

## 1. Introduction

Waspaloy superalloy is a γ′ phase precipitation reinforced nickel-based wrought superalloy [1,2]. The alloy has high strength and sufficient toughness at 650 °C~700 °C simultaneously, and is widely used in aerospace, petrochemical and other fields. In recent years, with the development of the performance stability of turbine discs for advanced aero-engines, the purity, grain size and strength of Waspaloy alloys are gradually demanding [3]. The melting process of Waspaloy has gradually changed from VIM (vacuum induction melting) to VIM + VAR (vacuum arc remelting), and then to VIM + ESR + VAR [4,5,6,7]. Among them, ESR provides high purity and high uniformity of electrode ingots for VAR, which is a vital part of metallurgical quality improvement [8,9]. It is essential to optimize ESR process parameters to achieve metallurgical quality improvement of Waspaloy.

However, a series of complex and coupled physical phenomena are involved in the process of ESR [10,11]. Moreover, the equipment, process and test measurement methods used in the whole smelting process are more complex. For large-scale industrial electroslag ingot production, the traditional trial and error method not only costs very much, but also many important internal operation parameters are difficult to obtain [12]. With the rapid development of computer application technology and the continuous progress of numerical simulation methods, the mathematical model simulation method can be used to more intuitively study the variation of molten pool morphology and solidification structure parameters with melting parameters in the ESR process [13,14,15,16]. DONG Yan-wu [16] established the mathematical models of electromagnetic field equation, fluid flow equation and temperature field for ESR, and studied the distribution of temperature field in the melting process. On this basis, a mathematical model coupled with magneto-hydrodynamics and a volume of fluid model was also established to study the melt droplet depression and inclusions behavior [17]. Qiang Wang [18] developed a transient three-dimensional (3D) model to investigate the role of filling rate in the macroscopic segregation process caused by convection of hot solutes from magnetic fluids during ESR. BAOKUAN LI [19] further developed a three-dimensional finite element model of the ESR process and investigated the electromagnetic field and liquid pool shape of the ESR process. Ebrahim Karimi-Sibaki [20] simulated and validated the industrial-scale ESR process, comparing the results of 3D calculations with 2D axisymmetric calculations, and concluded that 2D calculations are sufficient to solve for the ingot melt cell profile. The development of mathematical models for ESR has been relatively well developed. However, previous studies have mainly focused on the construction and refinement of the model, as well as the temperature field, electromagnetic field, solid–liquid transformation, etc. under a certain melting process. Changes in ESR process conditions cause changes in the temperature field, melt pool shape, etc., which in turn cause changes in the solidification organization. This has a huge impact on the metallurgical quality of the alloy. Solidification organization is a key characterization parameter of alloy quality, and it is meaningful to further develop the correlation model between melting process and solidification organization to explore the effect of process parameter changes on solidification organization. In this paper, Waspaloy with a diameter of 580 mm is studied. A 2D axisymmetric mathematical model has been developed by MeltFlow-ESR, which has been well-validated in the production of Inconel 718. The distribution of electromagnetic field, flow field, temperature field and melt pool morphology during ESR was discussed. The influence of electrode melting rate on dendrite spacing is compared, which provides reference for improving the quality of target products and optimizing process parameters.

## 2. Modelling

### 2.1. Model Description and Hypothesis

ESR process includes two stages: electrode melting and molten metal solidification in water-cooled mold. In the melting slag stage, due to the strong cooling effect of the cooling water tank at the bottom of the mold, in order to quickly establish the molten pool, the input power is usually increased to increase the Joule heat of the slag and make the electrode melt, then the power is gradually reduced to the range set by the process. In the stable smelting stage, the slag pool’s power input remains unchanged. With the growth of the solidified ingot, the influence of the cooling water tank at the bottom of the mold on its cooling effect gradually decreases, and the slag pool, molten pool, and solidified ingot maintain quasi-steady. In the stage of hot sealing, the power input into slag pool decreases gradually. Based on this, the calculation model of 2D axisymmetric process from unsteady state to steady state and then to unsteady state is established [17,19]. Due to the complex and coupled physical phenomena involved in ESR process, in order to reduce the difficulty of modeling and control the reasonable calculation time, the following assumptions are made for the model:(1)In the process of ESR, the interface between the electrode and slag pool and between the slag pool and the molten pool remains static and flat;(2)Neglecting the influence of droplet drop and slag movement on the flow field of the molten pool;(3)The thermophysical parameters of slag and alloy are considered as a function of temperature;(4)Because the influence of the electrode on and the magnetic field is limited to a very thin boundary layer, the calculation scope does not include the influence of electrode, that is, the insertion depth of electrode is not considered;(5)Regardless of the current flowing into the mold, it is considered that the slag skin formed in the ESR process has good electrical insulation;(6)The calculation range of the model starts from the top surface of slag, including slag pool, molten pool and a long enough ingot.

### 2.2. Electromagnetic Field

The electromagnetic phenomena of alternating current in ESR process conform to the magnetic quasi-static form controlled by Maxwell’s equations [20]. Under the condition of axial symmetry, the equations can be approximately simplified as the magnetic diffusion equation on the plane, as shown in Equation (1). By introducing the vector representation of alternating current to solve the equation, the distribution of magnetic field intensity ${\vec{H}}_{\theta }$ can be obtained. After further derivation, the current density $\vec{J}$ distribution, electromagnetic force $\vec{F}$ distribution, and Joule calorific value $S_{J}$ distribution are obtained, as shown in Equations (2)–(4):(1)$$\nabla \cdot \left(\frac{1}{\alpha }\nabla {\vec{H}}_{\theta }\right)=j\omega \mu_{0}{\vec{H}}_{\theta }$$

Current density:(2)$$\vec{J}=\nabla \times {\vec{H}}_{\theta }$$

Lorentz force:(3)$$\vec{F}=\vec{J}\times \vec{H}$$

Joule fever:(4)$$S_{J}=\frac{1}{2\alpha }\vec{J}\cdot \vec{J}$$
where ${\vec{H}}_{\theta }$ is the intensity of the magnetic field, A/m; $\vec{F}$ is the electromagnetic force, N/m^3^; $\vec{J}$ is the current density, A/m^2^; α is electrical conductivity, Ω^−1^·m^−1^; $\mu_{0}$ is the magnetic permeability, H/m; $S_{J}$ is Joule thermal power, W/m^3^.

Boundary Conditions: On the exposed surface of the slag and at the outer surface of the ingot, the value of the magnetic flux density is specified according to Ampere’s law in the following manner, as shown in Equations (5) and (6):

Exposed Slag Surface:(5)$${\vec{H}}_{\theta }\left(i\right)=\frac{1}{2\pi r}I$$

Ingot Circumferential Surface:(6)$${\vec{H}}_{\theta }\left(i\right)=\frac{1}{2\pi R_{\mathrm{Ingot}}}I$$
where ${\vec{H}}_{\theta }\left(i\right)$ is the axial intensity of the magnetic, A/m; r is the electrode radius, m; $R_{\mathrm{Ingot}}$ is the ingot radius, m; I is the total input current, A.

### 2.3. Flow Field

The fluid flow in slag pool and molten pool is mainly driven by buoyancy and electromagnetic force, and the flow is turbulent [21,22]. Therefore, the fluid flow in slag pool and molten pool can be controlled by continuity equation and Navier–Stokes equation, as shown in Equations (7) and (8), and the turbulent stirring degree can be calculated by RNG K–ε model, as shown in Equations (9) and (10):

Conservation equation of mass:(7)$$\frac{\partial \rho }{\partial t}+\nabla \cdot \left(\rho \vec{V}\right)=0$$

Momentum conservation equation:(8)$$\frac{\partial \left(\rho \vec{V}\right)}{\partial t}+\nabla \cdot \left(\rho V_{i}V_{j}\right)=-\nabla P+\nabla \cdot \left(\left(\mu_{l}+\mu_{t}\right)\left(\nabla V_{i}+\nabla V_{j}\right)\right)-\rho g\beta \left(T-T_{ref}\right)+{\vec{F}}_{L}$$

Turbulent kinetic energy:(9)$$\nabla \cdot \left(\rho \vec{V}k\right)=\nabla \cdot \left(\frac{\mu_{t}}{\sigma_{k}}\nabla k\right)+\mu_{t}G+\rho \varepsilon$$

Turbulent dissipation:(10)$$\nabla \cdot \left(\rho \vec{V}\varepsilon \right)=\nabla \cdot \left(\frac{\mu_{t}}{\sigma_{\varepsilon }}\nabla \varepsilon \right)+\left(c_{1}\mu_{t}G-c_{2}\rho \varepsilon \right)\frac{\varepsilon }{k}$$
where $\vec{V}$ is the velocity vector, $V_{i}$ is the axial velocity, $V_{j}$ is the radial velocity; P is pressure Pa; ${\vec{F}}_{L}$ is the external force source term, N; $\mu_{l}$ is laminar viscosity, Pa·s; $\mu_{t}$ is the turbulent viscosity, Pa·s.

Boundary Conditions: The no-slip boundary condition is imposed on the region of the mold boundary where the liquid phase (slag or molten metal) is in contact with the mold. The slag is subject to zero shear stress at the electrode-slag interface and the exposed surface of the slag, as shown in Equations (11) and (12).

Top Surface:(11)$$\partial \frac{V_{j}}{\partial i}=0$$

Mold Surface:(12)$$\vec{V_{}}=0$$

### 2.4. Temperature Field

The distribution of temperature field in slag pool and molten pool is mainly controlled by energy conservation equation, as shown in Equations (13) and (15), in which the external heat source considers the latent heat released during metal solidification and Joule heat generated by electromagnetic field, respectively.

Energy conservation equation:(13)$$\rho C_{P}\vec{V}\cdot \nabla T=\nabla \cdot \left(k\nabla T\right)-\rho \vec{V}L\nabla \cdot \lambda +S_{J}$$

Liquid fraction:(14)$$\lambda =f\left(\frac{T-T_{Solidus}}{T_{Liquidus}-T_{Solidus}}\right)$$

Further derivation:(15)$$\begin{array}{l} \frac{\partial \left(\rho \vec{V}C_{P}T\right)}{\partial t}+\nabla \left(\rho \vec{V}C_{P}T\right)=\nabla \cdot \left(k\nabla T\right)-\left(\frac{\partial \left(\rho \Delta H\right)}{\partial t}-\nabla \cdot \left(\rho \vec{V}\Delta H\right)\right)\\ -\left(\nabla \cdot \left(\rho \vec{V}\left(h-C_{P}T\right)\right)+\frac{\partial \left(\rho \vec{V}\left(h-C_{P}T\right)\right)}{\partial t}\right)+S_{J} \end{array}$$
where $C_{P}$ is the constant pressure heat capacity, kJ/(kg·K); $S_{J}$ is an external heat source.

Boundary Conditions: A uniform heat flux is prescribed at the slag-electrode interface. It corresponds to the energy required to heat the electrode from its inlet temperature to the liquidus temperature and any loss of heat that takes place from the exposed surface. The remaining surface of the slag is assumed to lose heat to the surroundings through radiation. The corresponding equations are described below:

Exposed Slag Surface:(16)$$-k\frac{\partial T}{\partial i}=\varepsilon \sigma \left(T^{4}-T_{\sin k}^{4}\right)$$
where σ is Stephen Boltzmann’s constant, 5.68 × 10^−8^ W/(m^2^·K^4^); ε is the emissivity of the slag.

### 2.5. Dendrite Spacing

When the ESR process enters the steady melting stage, the process parameters and the shape, volume and depth of the metal melting pool are basically stable, and no longer change with the smelting time. Therefore, the heating history, local solidification time and solidification structure parameters of solidified ingots are only determined by their position in ingots, where the dendrite spacing is calculated by Equations (17) and (18) [23,24]:(17)$$\lambda_{1}=\frac{150\times 10^{-6}}{{\left(G_{L}R\right)}^{0.33}}$$
(18)$$\lambda_{2}=\frac{40\times 10^{-6}}{{\left(G_{L}R\right)}^{0.42}}$$
where LST is the local solidification time, s; $\lambda_{1}$ is the primary dendrite spacing, m; $\lambda_{2}$ is the secondary dendrite spacing, m; $G_{L}R$ is the cooling rate, K·s^−1^; $G_{L}$ is the temperature gradient, K·m^−1^; R is the solidification rate, m·s^−1^.

### 2.6. Model Calculation Parameters

In this study, the electromagnetic field, flow field, temperature field and pool morphology of Waspaloy superalloy ESR system were simulated by MeltFlow-ESR Version 3.0 software, and the influence of melting rate on dendrite spacing was compared. Computational solution of the governing equations is carried out using the control volume method. The discretization equations for a physical variable represent exact conservation of the fluxes over each of the control volumes. The velocity-pressure coupling is handled using the SIMPLER algorithm. Since the discretization equations are nonlinear and coupled, iterations are required to attain convergence. The slag system used for the calculations is the CaF_2_-Al_2_O_3_-CaO-MgO-TiO_2_ five-member slag system. The physical parameters of the alloy and slag were mainly taken from the database that comes with Melt-Flow ESR software [25], and the melting parameters were taken from the Waspaloy melting process data provided by a superalloy production plant, as shown in Table 1 and Table 2.

## 3. Results and Discussions

### 3.1. Distribution of Temperature Field, Velocity Field and Molten Pool Morphology

Figure 1 shows the instantaneous distribution of temperature field, velocity field and molten pool morphology during ESR. It can be seen that the temperature in the slag pool is relatively high and the distribution is relatively uniform, and the temperature range is 1690–1830 K. The maximum temperature in the slag pool is 1729 K in the corner of the electrode tip, which is consistent with the distribution of the maximum current density and the maximum Joule thermal power. The minimum temperature appears at the interface of slag pool and mold wall, which is only 1587 K. The highest temperature of the molten pool appears at the center of slag–metal interface, which is 1686 K. Two pairs of circulating vortices exist in the slag cell, and the same flow field distribution is obtained for the three-dimensional ESR model developed by Ebrahim Karimi-Sibaki [25]. Near the inner wall of the mold, because the slag is cooled by the cooling water of the mold, the slag density in this area increases. Under the buoyancy [26], the slag flows clockwise along the mold wall, and the maximum flow velocity is 2.08 cm/s. In the center of the slag pool, due to the combined action of the inward and downward electromagnetic force and the falling liquid metal droplets, the slag flows counterclockwise and circularly down the center [27], and the maximum velocity is 3.05 cm/s.

The morphology of the metal bath is composed of a vertical section and the bottom of the parabola. The solidification point on the side of the ingot is maintained at a distance below the slag–metal interface, which ensures that the molten metal has a certain degree of superheat and is significantly higher than the melting temperature of the slag during the ascent of the molten pool, so the molten metal will come into contact with the solidified slag. At that time, part of the solidified slag will be re-melted, so that the slag will be thinly and evenly wrapped on the surface of the ingot. Due to the large amount of heat carried by the metal droplets through the slag pool, there is a certain degree of superheat in the metal melt pool. Under the cooling action of the mold cooling water, the melt pool area close to the mold wall has a fast heat transfer rate and a temperature gradient The molten pool is large, and the depth of the molten pool is shallow, so the entire molten pool presents the characteristics of a deep center and a shallow parabola around the molten pool and the shape of the molten pool is similar to “U” shape. Kanchan M. Kelkar showed that in the initial stage of the remelting process, the liquid metal pool has a very flat and shallow shape. As the ingot length increases, the curvature of the solidification front at the outer edge appears in a “U” shape due to the decrease of the heat extraction effect at the bottom, which is consistent with our results [25,28]. At the same time, the mushy zone also exhibits the characteristics of being farther from the center and narrower and narrower. The freezing point on the side of the molten pool is maintained at the slag-metal interface. The maximum molten pool depth is about 157 mm, and the maximum width of the mushy zone is about 135 mm.

### 3.2. The Influence of Electrode Diameter on Alloy Quality

#### 3.2.1. The Influence of Electrode Diameter on Temperature Field, Flow Field and Molten Pool Morphology

Figure 2 shows the temperature field, flow field and molten pool morphology changes of the slag pool and the metal molten pool under different filling conditions. It can be seen from the figure that the change of electrode diameter has a more impact on the temperature field of the slag pool. As the electrode diameter increases, the temperature distribution of the slag pool becomes more uniform. When the diameter of the filling electrode is 440 mm and 500 mm, the average temperature is 1698 K and 1703 K, respectively, and the electrode diameter has little effect on the temperature field and the shape of the molten pool. Therefore, appropriately increasing the electrode diameter under the premise of ensuring safety is more conducive to the uniformity of the reaction between molten slag and molten metal.

#### 3.2.2. The Influence of Electrode Diameter on Dendrite Spacing

Figure 3 show the changes of the primary dendrite spacing and the secondary dendrite spacing under different electrode diameter conditions. It can be seen from the figure that the change of electrode diameter has little effect on the solidification structure of the remelted ingot. With increasing electrode diameter, the same section height, primary dendrite secondary dendrite arm spacing and the same trend, the size of the dendrite arm spacing or less.

### 3.3. Study on the Influence of Melting Rate on the Morphology of Molten Pool

#### 3.3.1. The Effect of Melting Rate on Temperature Field, Flow Field and Molten Pool Morphology

Figure 4 and Figure 5, respectively, show the temperature field, flow field and molten pool morphology changes of the slag pool and the metal molten pool under different melting rate conditions. It can be seen from the figure that the melting rate has a greater influence on the temperature field of the slag pool and the molten metal pool. As the melting rate increases, the temperature of the slag pool increases, and the high temperature zone continues to expand. When the melting rate is 5.4 kg/min, the high temperature zone is mainly concentrated in the corner area of the lower end of the electrode. About 1670 K; when the melting rate increases to 6.4 kg/min, the high temperature zone is evenly distributed in the slag pool, the highest temperature is about 1721 K, and the temperature at the slag-metal interface is about 1699 K. In addition, as the melting rate increases, the amount of heat entering the molten metal pool per unit time increases, and the temperature in the molten pool rises accordingly. Further analysis found that as the melting rate increases, the volume of the molten metal pool continues to increase, the depth of the molten pool and the width of the mushy zone also increase correspondingly [29], and the overall morphology of the metal molten pool gradually changes from “U” to “V”. When the melting rate is 5.4 kg/min, the maximum depth of the molten pool is about 200 mm, and the maximum width of the mushy zone is about 100 mm; when the melting rate is increased to 6.4 kg/min, the maximum depth of the molten pool is about 260 mm, and the maximum mushy zone width is about 120 mm.

#### 3.3.2. Study on the Effect of Melting Rate on Dendrite Spacing

Relative to the electrode diameter, the variation of melting rate has more influence on the average temperature and solidification structure of remelted ingots, as shown in Table 3. When the melting rate was 5.4 kg/min, the maximum primary dendrite spacing of the ingot was about 680.99 μm, and when the melting rate was increased to 6.4 kg/min, the maximum primary dendrite spacing of the ingot was about 796.14 μm. Due to the small adjustable range of the electrode diameter and its limited influence, optimization of the melting rate is the key factor to improve the solidification conditions of the ingot.

Figure 6 and Figure 7 show the variations of the primary dendrite spacing and secondary dendrite spacing for different melting rate conditions. From the figures, it can be seen that the primary and secondary dendrite spacing of the ingot tends to increase with the increase of melting rate. The optimum melting rate is 5.8–6.0 kg/min for the combined uniformity of the dendrite distribution.

## 4. Conclusions

Transient numerical simulations were used to study the temperature, velocity and solidification fields of a 580 mm diameter Waspaloy superalloy during ESR. The effects of melting rate, filling rate, and thickness of the slag layer on the molten pool shape together with dendrite arm spacing evolution have been discussed.
(1)The temperature in the slag pool is high and relatively uniformly distributed, with temperature variations ranging from 1690 to 1830 K. There are two pairs of circulating vortices in the slag pool, the buoyancy of the slag causes the side vortices due to density difference, and the center vortices are the joint action of electromagnetic force and momentum of falling metal droplets. The whole melt pool shows the parabolic characteristics of deep center and shallow surroundings, and the shape of the melt pool is similar to “U”. The highest temperature of the molten pool appears in the center of the slag–metal interface, 1686 K.(2)The increase in electrode diameter has little effect on the solidification structure of the molten pool and ingot; however, it can significantly improve the uniformity and fluidity of the slag pool temperature, which is conducive to the full contact and reaction of liquid metal and molten slag to remove inclusions and harmful impurities. Under the premise of ensuring safety, the electrode diameter can be appropriately increased.(3)If the influence of droplet on the flow field of molten pool is ignored. As the melting rate increases, the slag pool and the high temperature zone of the molten pool gradually increase, the volume of the molten pool and the mushy zone gradually increase, and the depth and width of the mushy zone increase and gradually become narrower. The overall shape of the melt pool changes from “U” to “V” shape, and the overall distribution of the dendrite spacing changes from uneven to uniform to uneven. Considering the distribution of melt pool shape and dendrite spacing, the melting rate should be kept at 5.8 kg/min.

## Figures and Tables

**Figure 1 materials-15-07483-f001:**
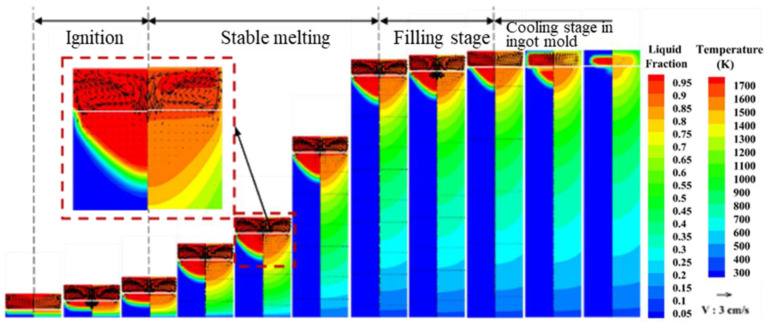
Temperature field, velocity field and molten pool morphology distribution.

**Figure 2 materials-15-07483-f002:**
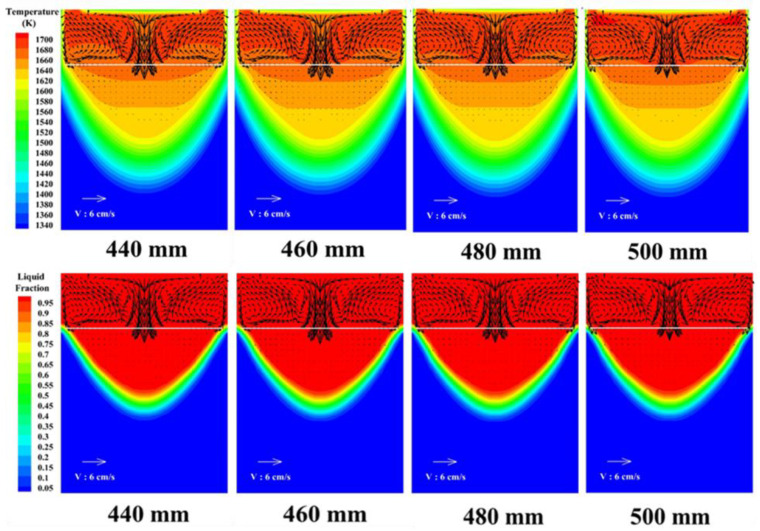
The influence of electrode diameter on temperature field, flow field distribution and molten pool distribution.

**Figure 3 materials-15-07483-f003:**
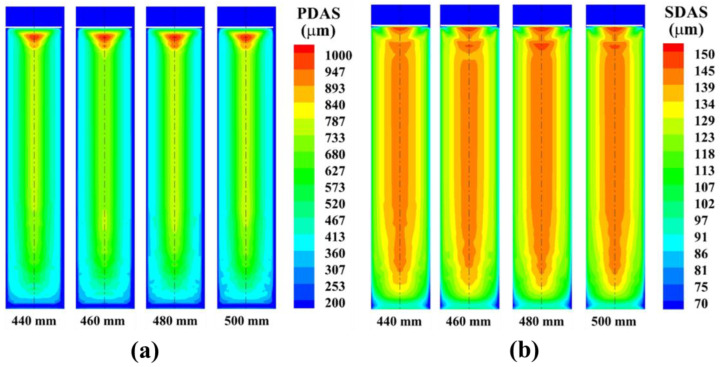
The influence of electrode diameter on the distribution of (**a**) primary and (**b**) secondary dendrite spacing.

**Figure 4 materials-15-07483-f004:**
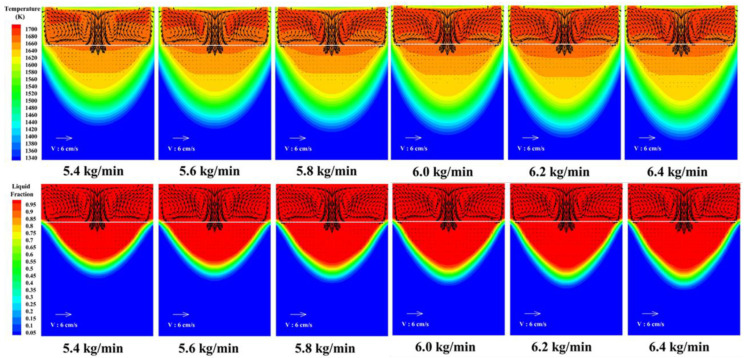
Changes in temperature field, velocity field and molten pool morphology.

**Figure 5 materials-15-07483-f005:**
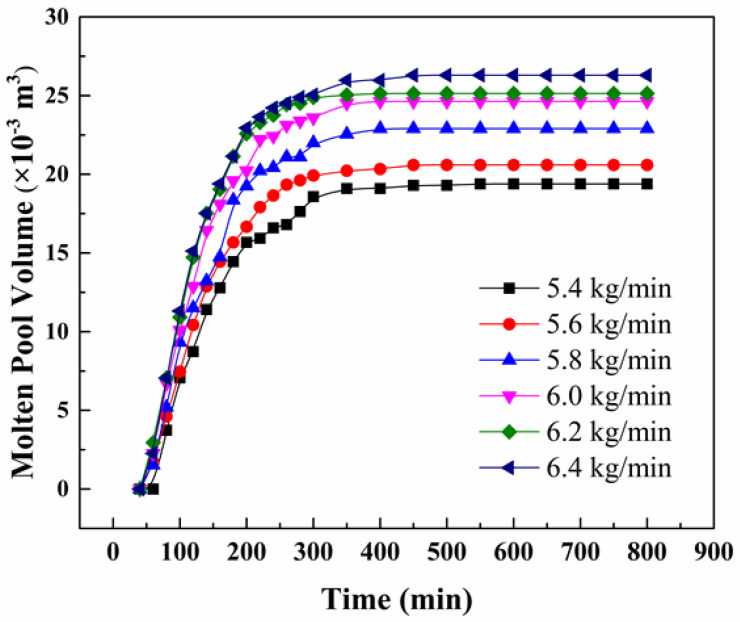
Changes in molten pool volume and depth.

**Figure 6 materials-15-07483-f006:**
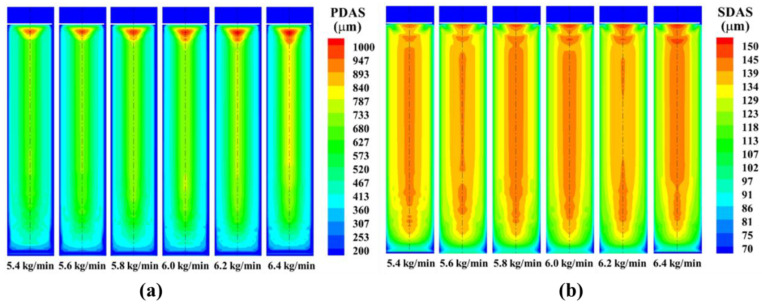
The influence of melting rate on the distribution of (**a**) primary and (**b**) secondary dendrite spacing.

**Figure 7 materials-15-07483-f007:**
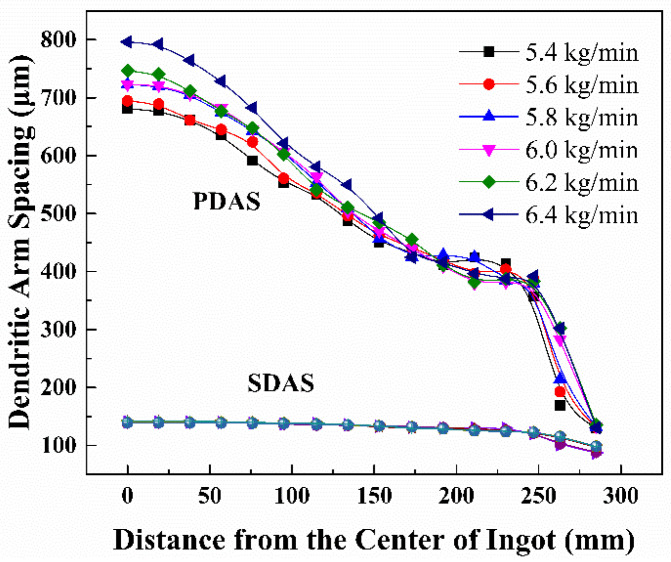
The effect of melting rate on the size of dendrite spacing.

**Table 1 materials-15-07483-t001:** Physical parameters of slag and alloy.

Parameters	Latent Heat, J/kg	Conductivity, Amp/V·m	Density, kg/m^3^	Viscosity, Pa·s	Solid–Liquid Phase Line Temperature, K
Waspaloy	2.4 × 10^5^	10.20 × 10^5^	8230	7.49 × 10^−3^~5.64 × 10^−3^	1483~1634
Slag	4.3 × 10^5^	363	2630	3.26 × 10^−2^~1.93 × 10^−2^	1473~1553

**Table 2 materials-15-07483-t002:** Process parameters of the ESR.

Ingot Diameter, mm	Current, kA	Slag Height, mm	Electrode Diameter, mm
580	8.4	150	440, 460, 480, 500
Frequency, Hz	Cooling water flow, L/min	Cooling water flow, L/min	Melting speed, kg/min
50	309.5	920	5.4, 5.6, 5.8, 6.0, 6.2, 6.4

**Table 3 materials-15-07483-t003:** Average temperature and solidification structure of remelted ingot under different process conditions.

Typical Result	Electrode Diameter		Melting Rate	
	440 mm	500 mm	5.4 kg/min	6.4 kg/min
Average temperature (K)	1689	1703	1670	1721
Maximum primary dendrite spacing (μm)	767.25	749.39	680.99	796.14

## Data Availability

Not applicable.

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
