# Peer review of "Optimization of Process Parameters for ESR Waspaloy Superalloy by Numerical Simulation"

_materials, 2022, doi:10.3390/ma15217483_

Round 1
Reviewer 1 Report
Dear Authors,
Major remarks:
1. The introduction requires improvement – motivation, aim, novelty, and literature review should be improved.
2. Table 1 – please provide references for the used parameters.
3. Lines 146-147 “the electromagnetic field, flow field, temperature field and pool morphology of Waspaloy superalloy ESR system were simulated by Melt-Flow ESR software”- Please provide more information solver, modeling process and about numerical model/boundary conditions/etc.
4. Some figures (1, 2, 4) are blurred and should be improved
5. The description of the results should be improved. At the moment, section 3 seems like a technical report. The crucial results of the analyses should be set in tables. What about comparison ad discussion of the obtained results with other studies?
6. Designation of figures (a and b) should be below figures (Figures 3,6).
7. Lines 240-245 “The specific performance is: when the melting rate is 5.4 kg/min, the maximum…..” – grammar /style issues.
8. What about the verification of the numerical solutions? Has the modeling technique been verified with the experiment, similar experimental results from the literature, or by solution obtained by another software? Were the convergence tests performed?
Minor remarks:
1. Line 24 “ESR process” – please add full name in the Abstract.
2. Line 40 “electroslag remelting [10, 11] , Moreover, the equipment, process” – please correct comma.
3. Line 55/56 “The influence of electrode melting rate on electroslag remelting process is compared”
and
Line 148 “and the influence of melting rate on dendrite spacing was compared.”– compared with what? Please clarify.
4. Line 138 “where the dendrite spacing is calculated by the equation [23, 24] as shown in equations (12)-(13)” – please correct
Kind Regards,
Reviewer 2 Report
Although the authors presented new results through simulation, there is some margin of error between simulation and experimental results. Therefore, when presenting new information by simulation, it is common to report results that have been discussed while comparing them with experimental results. As the author explains, some assumptions have been proposed to reduce the complexity of the simulation process, but some of them are not realistic. For example, neglecting the influence of droplet drop and slag movement on the flow field of molten pool...
Consequently, authors should obtain experimental results to improve the reliability of the reported results, and further discussion is urgently needed.
Round 2
Reviewer 1 Report
Dear Authors,
Remarks:
1. Fonts should be unified, mainly in units and equations.
2. Grammar and style should be carefully revised. In the example, in section 2.6 “Computational solution of the governing equations is carried using the control volume method.” in my opinion, it should be “carried out” instead “carried”. A few times two dots occurs at the end of phrases.
3. Still, the motivation of the study is not sufficiently justified. “However, fewer have studied the correlation between process parameters and solidification organization to provide guidance for industrial production.” This problem should be more precisely described, and the novelty of the study should be emphasized with respect to the method available in the literature.
Kind Regards.
Reviewer 2 Report
The answers are acceptable.
